# Hy-ClustRec: Deep Cluster-Guided Rule Mining for Cold-Start Recommendation

## Abstract

Recommender systems are critical for navigating vast item catalogs but struggle with the cold-start problem, where a lack of interaction data degrades recommendation quality. While hybrid methods exist, they often fail to effectively structure item content or extract fine-grained preference patterns. In this paper, we introduce **Hy-ClustRec**, a novel three-stage framework designed to address these challenges. First, we learn dense, non-linear representations of item content using a deep autoencoder. Second, these embeddings are segmented into meaningful communities using HDBSCAN, a density-based clustering algorithm. Third, we employ a hierarchical strategy for Association Rule Mining (ARM) to discover global and specialized co-occurrence patterns. Candidate items are then ranked using a hybrid scoring function that fuses rule confidence, semantic similarity from SBERT embeddings, and a user-cluster affinity score. To further boost performance, we incorporate an item-based KNN recommender into the final score with a weighted sum. Evaluated on a sparse subset of the Million Song Dataset, Hy-ClustRec demonstrates strong performance, proving especially effective in cold-start scenarios. Our work shows that a structured pipeline combining deep clustering with hierarchical rule mining and collaborative signals offers a robust solution to the cold-start problem.

## 1 Introduction

Recommender systems (RS) are fundamental to modern digital platforms (Ricci et al., 2011), but their effectiveness hinges on learning from historical user-item interactions. This dependency creates a critical vulnerability known as the **cold-start problem**: the inability to provide relevant recommendations for new users or items lacking interaction data (Khaledian et al., 2025; Kulkarni et al., 2021). Solving this is a central requirement for sustained user engagement on platforms that constantly onboard new content and users (Al-Obeidi et al., 2018). The cold-start scenario is the most severe form of the broader **data sparsity** issue, where even established entities have a tiny footprint in the user-item matrix, making robust representation learning difficult for traditional collaborative models (Su & Khoshgoftaar, 2009; Roy & Dutta, 2022; Ahmadian et al., 2022).

Two dominant paradigms have emerged to address this challenge. The first, **content-alignment**, involves hybrid models that leverage rich item metadata to inform collaborative embeddings, often by aligning or fusing representations from multiple modalities into a shared space (Ganhör et al., 2024; Wei et al., 2021). While powerful, these methods often operate as complex, end-to-end models, making it difficult to disentangle the effects of content structure from collaborative patterns (Park et al., 2022). The second paradigm relies on **structured pattern mining**, employing a transparent pipeline that first clusters the item space and then applies Association Rule Mining (ARM) to discover co-occurrence patterns within these denser groups (Najafabadi et al., 2017; Khaledian et al., 2025). The strength of this approach lies in its interpretability; however, it typically relies on shallow, raw content features, failing to capture deeper, non-linear semantic relationships that deep learning excels at. This reveals a research gap: Can we unite the *deep representation learning* of content-alignment models with the *principled, hierarchical pattern mining* of structured pipelines?

In this paper, we introduce **Hy-ClustRec**, a novel three-stage model that leverages deep learning not as an end-to-end predictor, but as a powerful feature engineering step to create a semantically rich foundation for a structured and interpretable pattern mining process. Our contributions are:

1. **Deep Content-Based Clustering:** We train a deep **autoencoder** to learn dense, non-linear representations of item metadata, then segment items into meaningful communities using **HDBSCAN**, a density-based algorithm that uncovers arbitrarily shaped clusters and identifies outliers.

2. **Hierarchical Association Rule Mining (ARM):** We mine **intra-cluster rules** for fine-grained, community-specific patterns and **global rules** for broad popularity trends, generating a comprehensive and diverse set of candidate items.

3. **Semantic and Affinity-Based Re-ranking:** We generate recommendations using a hybrid scoring function that fuses rule **confidence**, SBERT-based **semantic similarity**, and a novel **user-cluster affinity score**. Additionally, we incorporate an **item-based KNN** component (Sarwar et al., 2001) into the final scoring via a weighted combination.

This structured pipeline offers a unique advantage by using deep learning to enhance a transparent pattern-mining framework, providing a robust and methodologically novel solution for cold-start recommendation.

## 2    RELATED WORK

Our work, Hy-ClustRec, builds upon two primary research paradigms: structured pattern mining and deep content-based recommendation.

### 2.1    HYBRID AND CONTENT-AWARE MODELS FOR COLD-START

A common cold-start strategy is creating hybrid models that leverage item content when interaction data is sparse (Kulkarni et al., 2021; Chen et al., 2019; Al-Obeidi et al., 2018). Early work in music recommendation used deep convolutional networks directly on audio content to learn effective representations (Van den Oord et al., 2013), and models like Variational Autoencoders have shown promise in modeling user preferences from implicit feedback (Liang et al., 2018). Other prominent methods use content to regularize collaborative models. **DropoutNet** applies dropout to input feature vectors, forcing the model to predict with missing data (Volkovs et al., 2017), while **MARec** aligns a collaborative model with a metadata-derived similarity matrix (Monteil et al., 2024). Traditional item-based collaborative filtering (ItemKNN) is another classical method, computing item-item similarities from interaction data (Sarwar et al., 2001), but without content it struggles in cold-start scenarios. Hy-ClustRec differs by using deep learning in a modular, pipeline-based fashion, where learned content representations serve as high-quality inputs for a distinct, structured pattern mining stage.

### 2.2    CLUSTERING AND ASSOCIATION RULE MINING FOR RECOMMENDATION

A parallel line of work uses a pipeline of unsupervised techniques to mitigate data sparsity by clustering items or users and then applying ARM to discover co-occurrence patterns within these denser groups (Najafabadi et al., 2017). **CFCAI**, for example, recently demonstrated this pipeline's effectiveness on sparse data by combining item clustering with ARM (Khaledian et al., 2025). While structurally similar, our work advances this paradigm by: (1) using a **deep autoencoder** to learn non-linear representations for clustering, (2) employing the more robust **HDBSCAN** algorithm to find arbitrarily shaped clusters and identify noise, and (3) introducing a **hierarchical ARM** strategy that mines both fine-grained intra-cluster rules and broad global rules.

### 2.3    ADVANCED APPROACHES AND FUTURE DIRECTIONS

Recent research has pushed the boundaries of cold-start recommendation. Meta-learning frameworks learn adaptable preference estimators that generalize from few interactions (Lee et al., 2019), while Graph Neural Networks (GNNs) model the user-item graph to capture high-order collaborative signals (He et al., 2020). Other works tackle the extreme "zero-shot" scenario (Ding et al., 2021) or use Large Language Models (LLMs) for conversational and controllable recommendation (Dong et al., 2024; Carroll et al., 2025). While Hy-ClustRec does not rely on meta-learning or GNNs, its content-centric approach allows it to handle "zero-interaction" items, effectively performing a type of zero-shot recommendation within a single domain.

# 3 PROPOSED METHOD

We detail the three-phase architecture of **Hy-ClustRec**, a framework designed to address the cold-start problem by combining deep representation learning with structured pattern mining. A conceptual overview is provided in Figure 1.

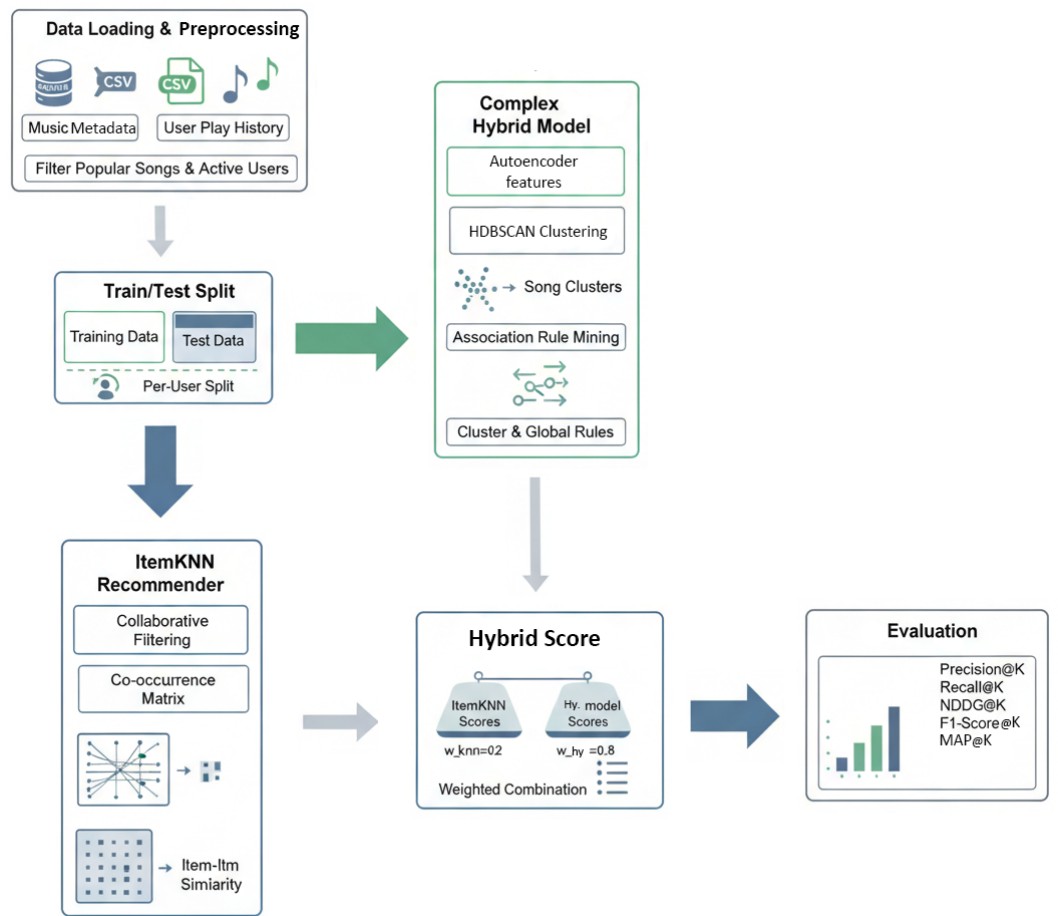

Figure 1: Conceptual overview of the Hy-ClustRec framework.

## 3.1 PROBLEM FORMULATION

Given a set of users $\mathcal{U}$ and items $\mathcal{I}$, user-item interactions are represented by a sparse matrix $\boldsymbol{R}$, where $R_{ui} = 1$ indicates an implicit interaction. Each item $i \in \mathcal{I}$ has content metadata. The goal is to generate a ranked list of top-$N$ recommendations for each user, particularly for those with a sparse interaction history.

## 3.2 PHASE 1: DEEP CONTENT-BASED ITEM CLUSTERING

### 3.2.1 DEEP CONTENT REPRESENTATION WITH AN AUTOENCODER

To learn a dense and non-linear representation of item content, we use a deep **autoencoder** (Sedhain et al., 2015). An item's standardized metadata vector $\boldsymbol{f}'_i$ is fed into an autoencoder with an encoder $f_{\text{enc}}$ and decoder $f_{\text{dec}}$, trained to minimize the reconstruction error:

$$\mathcal{L}_{AE} = \sum_{i \in \mathcal{I}} \|\boldsymbol{f}'_i - f_{\text{dec}}(f_{\text{enc}}(\boldsymbol{f}'_i))\|_2^2.$$

After training, the encoder produces a low-dimensional content representation for each item:

$$\boldsymbol{v}_i^{\text{content}} = f_{\text{enc}}(\boldsymbol{f}_i') \in \mathbb{R}^{d_c}. \tag{1}$$

### 3.2.2 DENSITY-BASED ITEM SEGMENTATION WITH HDBSCAN

Using the learned representations $\{\boldsymbol{v}_i^{\text{content}}\}_{i\in\mathcal{I}}$, we segment items with the **HDBSCAN** algorithm (Campello et al., 2013). Unlike $k$-Means, HDBSCAN can find arbitrarily shaped clusters and identify noise without a predefined cluster count. This partitions the item set $\mathcal{I}$ into $k$ disjoint clusters $C_1, \ldots, C_k$ and a set of noise points $C_{\text{noise}}$.

## 3.3 PHASE 2: HIERARCHICAL ASSOCIATION RULE MINING (ARM)

We generate recommendation candidates by mining co-occurrence patterns using the efficient **FP-Growth** algorithm (Han et al., 2000). We evaluate rules of the form $\{A\} \rightarrow \{B\}$ using standard metrics like Support, Confidence, and Lift. For an itemset $X$ and the set of user histories $\mathcal{H}$:

$$\text{Support}(X) = \frac{|\{h \in \mathcal{H} \mid X \subseteq h\}|}{|\mathcal{H}|}.$$

The rule metrics are defined as:

$$\text{Confidence}(A \rightarrow B) = \frac{\text{Support}(A \cup B)}{\text{Support}(A)}, \quad \text{Lift}(A \rightarrow B) = \frac{\text{Confidence}(A \rightarrow B)}{\text{Support}(B)}. \tag{2}$$

We mine two sets of rules: **Intra-Cluster Rules** from data within each cluster to find community-specific patterns, and **Global Rules** from the entire dataset to capture broad popularity trends.

## 3.4 PHASE 3: HYBRID SCORING AND RE-RANKING

The final phase scores and ranks candidate items using a hybrid function that fuses three signals:

1. **Rule-Based Confidence (Conf$_{\text{norm}}$):** The confidence of the ARM rule, reflecting collaborative strength.

2. **Semantic Content Similarity (Sim$_{\text{norm}}$):** The cosine similarity between a candidate's SBERT embedding (Reimers & Gurevych, 2019) and the user's average profile embedding. Let $\boldsymbol{v}_s$ be the SBERT vector for candidate $s$ and $\bar{\boldsymbol{v}}_u$ be the mean SBERT vector of items in user $u$'s history. The similarity is:

$$\text{Sim}(s, u) = \frac{\boldsymbol{v}_s \cdot \bar{\boldsymbol{v}}_u}{\|\boldsymbol{v}_s\| \|\bar{\boldsymbol{v}}_u\|}.$$

3. **User-Cluster Membership (Mem$_{\text{norm}}$):** A user's affinity for an item's cluster, calculated as the proportion of a user's listening history belonging to that cluster:

$$\text{Membership}(u, C_j) = \frac{\sum_{i \in C_j} R_{ui} \cdot \text{play\_count}_{ui}}{\sum_{i \in \mathcal{I}} R_{ui} \cdot \text{play\_count}_{ui}}. \tag{3}$$

This score indicates how much a user's taste aligns with the content community of a candidate item.

These three signals are normalized to a $[0, 1]$ range using Min-Max scaling. The final score for a candidate item $s$ for user $u$ is a weighted sum:

$$\text{Score}_{\text{Hy}}(u, s) = w_c \cdot \text{Conf}_{\text{norm}} + w_s \cdot \text{Sim}_{\text{norm}} + w_m \cdot \text{Mem}_{\text{norm}}, \tag{4}$$

where weights $w_c, w_s, w_m$ are tuned on a validation set to balance the contributions of each component. In our experiments, for example, we set $w_c = 4.5$, $w_s = 0.5$, and $w_m = 0.5$ to emphasize the rule confidence.

### 3.4.1 ITEM-BASED KNN HYBRID SCORING

We augment Hy-ClustRec with an **item-based KNN** component (Sarwar et al., 2001). Instead of cosine/Pearson over explicit ratings, we use a co-occurrence similarity tailored to sparse implicit data: let $M \in \{0,1\}^{|\mathcal{I}| \times |\mathcal{U}|}$ be the item×user matrix ($M_{i,u} = 1$ iff user $u$ played item $i$); we form $S = MM^\top$ with $s_{ij}$ the number of co-listeners and set $s_{ii} = 0$,

$$S = MM^\top, \quad s_{ii} = 0. \tag{5}$$

For user $u$ and candidate $i$, we sum similarities to *all* history items (no $k$-truncation): $\hat{r}_{ui}^{(\mathrm{KNN})} = \sum_{j \in \mathcal{I}_u} s_{ij}$; cold-start users back off to a MostPopular list. Because score scales differ, we apply per-user MinMax normalization to each model's scores and fuse them with an 80/20 weighting:

$$\mathrm{Score}_{\mathrm{final}}(u,i) = 0.8\,\mathrm{norm}_u\big(\mathrm{Score}_{\mathrm{Hy}}(u,i)\big) + 0.2\,\mathrm{norm}_u\big(\hat{r}_{ui}^{(\mathrm{KNN})}\big). \tag{6}$$

This matches our implementation while remaining consistent with the classical item-based CF view of Sarwar et al. (2001).

## 4 EXPERIMENTS

### 4.1 DATASET AND PREPROCESSING

We use the **Million Song Dataset (MSD)** (Dataset, 2011), augmented with metadata for genre and artist information. To create a challenging yet manageable sparse environment, we sample 30% of users, then filter to retain only songs played at least 70 times and users with at least 20 interactions. The final dataset presents a significant cold-start challenge.

### 4.2 EXPERIMENTAL SETUP

**Data Splitting.** For each user with $\geq 20$ interactions, we randomly partition their history into an **80% training set** and a **20% test set**. The split is performed independently for each user, ensuring both sets contain only items the user has interacted with.

**Evaluation Metrics.** We evaluate top-$N$ recommendations using standard ranking metrics for $K \in \{1, 5, 10, 20, 30\}$. Key metrics include Precision@K, Recall@K, F1-score@K, NDCG@K, and MAP@K. For a user's ranked list of recommendations $R_u^K$ and their ground-truth test items $T_u$:

$$\mathrm{Precision@K} = \frac{|R_u^K \cap T_u|}{K}, \quad \mathrm{Recall@K} = \frac{|R_u^K \cap T_u|}{|T_u|}. \tag{7}$$

$$\mathrm{F1@K} = \frac{2 \cdot \mathrm{Precision@K} \cdot \mathrm{Recall@K}}{\mathrm{Precision@K} + \mathrm{Recall@K}} \tag{8}$$

We also compute NDCG@K by:

$$\mathrm{DCG@K} = \sum_{i=1}^{K} \frac{rel_i}{\log_2(i+1)}, \quad \mathrm{NDCG@K} = \frac{\mathrm{DCG@K}}{\mathrm{IDCG@K}}, \tag{9}$$

where $rel_i = 1$ if the item at rank $i$ is relevant (in $T_u$), and 0 otherwise. IDCG@K is the DCG of the ideal ranking. F1@K is the harmonic mean of Precision@K and Recall@K, and MAP@K is the mean average precision up to $K$. For a user's ranked list of recommendations $R_u^K$ and their ground-truth test items $T_u$, the Average Precision (AP) is:

$$AP@K = \frac{1}{\min(|T_u|, K)} \sum_{i=1}^{K} P@i \cdot rel_i \tag{10}$$

where $P@i$ is the precision at cutoff $i$, and $rel_i = 1$ if the item at rank $i$ is relevant, and 0 otherwise. The Mean Average Precision (MAP) is then obtained by averaging AP over all users:

$$MAP@K = \frac{1}{|\mathcal{U}|} \sum_{u \in \mathcal{U}} AP@K(u). \tag{11}$$

**Baselines.** We compare Hy-ClustRec with two baseline methods: **ItemKNN** and **CFCAI**. The ItemKNN approach is an item-based $k$-Nearest Neighbors collaborative filtering method (Sarwar et al., 2001), which recommends items similar to those in the user's history based on item-item co-occurrence. The CFCAI method is a recent cluster-based recommendation pipeline (Khaledian et al., 2025) that also uses item clustering and rule mining; we run CFCAI on our dataset for a direct comparison. These baselines represent standard collaborative filtering strategies and a state-of-the-art clustering+ARM approach, respectively.

## 4.3 PERFORMANCE OF HY-CLUSTREC

Table 1 summarizes the performance of our proposed model. Hy-ClustRec demonstrates strong performance even in this highly sparse setting. For example, it achieves a **Precision@1 of 25.12%** and an **NDCG@10 of 21.51%**, substantially outperforming the baselines on ranking quality. These results highlight the model's ability to accurately rank relevant items with limited interaction data, benefitting from its structured pipeline that successfully integrates deep content understanding with collaborative pattern mining.

Table 1: Top-$K$ recommendation performance of **Hy-ClustRec** (percent).

| K | Precision@K | Recall@K | F1-score@K | MAP@K | NDCG@K |
|---|---|---|---|---|---|
| 1 | 25.12% | 3.56% | 6.11% | 25.12% | 25.37% |
| 5 | 17.56% | 12.43% | 14.65% | 12.11% | 20.25% |
| 10 | 13.98% | 18.92% | 15.16% | 11.11% | 21.51% |
| 20 | 10.41% | 27.49% | 14.28% | 11.85% | 23.55% |
| 30 | 8.86% | 34.20% | 13.35% | 12.66% | 26.21% |

Table 2 compares Hy-ClustRec against the two baselines. Hy-ClustRec achieves substantially higher NDCG across all $K$ values. For instance, at $K = 10$, Hy-ClustRec's NDCG is 0.2151, compared to 0.2040 for ItemKNN and 0.1064 for CFCAI. This indicates that Hy-ClustRec ranks relevant items significantly higher in the recommended list. The ItemKNN baseline tends to retrieve a larger number of relevant items (higher recall) by casting a wider net, whereas CFCAI often struggles with precision. In contrast, Hy-ClustRec strikes a balance: it maintains high ranking quality (NDCG) and precision while achieving competitive recall.

Table 2: Comparison of **Hy-ClustRec** with baselines on Top-$K$ recommendation (percent).

| Metric | Model | @1 | @5 | @10 | @20 | @30 |
|---|---|---|---|---|---|---|
| Precision | ItemKNN | **25.20%** | **18.30%** | **14.00%** | 10.40% | 8.70% |
| | CFCAI | 14.27% | 9.32% | 6.56% | 4.66% | 3.72% |
| | **Hy-ClustRec** | 25.12% | 17.56% | 13.98% | **10.41%** | **8.86%** |
| Recall | ItemKNN | 3.50% | 12.40% | 18.50% | 26.70% | 33.30% |
| | CFCAI | 1.85% | 5.93% | 8.17% | 11.77% | 14.21% |
| | **Hy-ClustRec** | **3.56%** | **12.43%** | **18.92%** | **27.49%** | **34.20%** |
| F1-score | ItemKNN | 6.00% | 14.00% | 14.90% | 14.00% | 13.10% |
| | CFCAI | 3.20% | 6.84% | 6.80% | 6.30% | 5.60% |
| | **Hy-ClustRec** | **6.11%** | **14.56%** | **15.16%** | **14.28%** | **13.35%** |
| NDCG | ItemKNN | 25.20% | 20.00% | 20.40% | 23.30% | 25.90% |
| | CFCAI | 14.27% | 10.94% | 10.64% | 11.83% | 12.80% |
| | **Hy-ClustRec** | **25.37%** | **20.25%** | **21.51%** | **23.55%** | **26.21%** |
| MAP | ItemKNN | 3.50% | 8.40% | 10.40% | **12.00%** | **12.90%** |
| | CFCAI | 14.27% | 6.53% | 5.26% | 5.23% | 5.36% |
| | **Hy-ClustRec** | **25.12%** | **12.11%** | **11.11%** | 11.85% | 12.66% |

## 4.4 Ablation Study

To validate the contribution of each component in our framework, we performed an ablation study (results at $K = 10$ are shown in Table 3). We evaluated the full model against several variants: clustering on raw features instead of autoencoder embeddings (*w/o Autoencoder*), replacing HDB-SCAN with K-Means (*w/ K-Means*), and removing the semantic similarity (*w/o SBERT Similarity*) or user-cluster affinity (*w/o Cluster Affinity*) from the final scoring function. The results quantify the performance impact of each component and confirm their synergistic contributions.

Table 3: Ablation study results for Hy-ClustRec at $K = 10$.

| Model Variant | Precision@10 | Recall@10 | NDCG@10 |
|---|---|---|---|
| **Hy-ClustRec (Full Model)** | **13.98%** | **18.92%** | **21.51%** |
| *w/o Autoencoder* | 12.02% | 15.57% | 18.04% |
| *w/ K-Means* | 12.29% | 16.08% | 18.37% |
| *w/o SBERT Similarity* | 12.00% | 15.51% | 17.86% |
| *w/o Cluster Affinity* | 12.05% | 15.53% | 17.85% |

**Analysis of Ablation Results.** The full Hy-ClustRec model consistently outperforms all ablated variants, demonstrating that each component contributes positively to the recommendation quality. Removing the autoencoder leads to a noticeable drop in performance (Precision falls from 13.98% to 12.02% at $K = 10$), highlighting the importance of learning a dense, non-linear content representation. Replacing HDBSCAN with K-Means degrades performance further, justifying our choice of a robust density-based clustering algorithm. Finally, removing either the SBERT similarity or the cluster affinity score also reduces performance, confirming that both signals complement the primary collaborative signal from rule confidence.

## 5 Conclusion

In this paper, we introduced **Hy-ClustRec**, a novel three-stage hybrid framework that addresses the cold-start problem by combining deep content representation learning, hierarchical association rule mining, and a fused scoring function. Our method first employs a deep autoencoder and HDB-SCAN to create semantically coherent item communities. It then extracts both intra-cluster and global collaborative signals via hierarchical ARM. Finally, a unique hybrid scoring function re-ranks candidates by fusing rule-based confidence, SBERT-derived semantic similarity, and user-cluster affinity. As an additional boost, we incorporate an item-based KNN component following Sarwar et al. (2001). Experiments on a sparse subset of the Million Song Dataset demonstrate that Hy-ClustRec achieves strong performance, accurately identifying relevant items even in cold-start settings. A thorough ablation study further validated the positive contribution of each component in our integrated pipeline.

For future work, our framework could be extended by incorporating richer content modalities and exploring adaptive weighting mechanisms for the scoring function. Moreover, investigating potential biases in the item metadata and learned representations to ensure fairness is an important direction. In conclusion, Hy-ClustRec offers a robust and transparent solution to the cold-start problem, providing valuable insights for developing more intelligent and adaptive recommender systems.

## Ethics Statement

We adhere to the ICLR Code of Ethics. This work uses the publicly available Million Song Dataset. While our model aims to improve recommendation quality, content-based methods may inadvertently amplify biases present in the underlying data (e.g., historical overrepresentation of certain genres). We recognize the importance of fairness and plan to investigate methods for bias detection and mitigation, such as fairness-aware clustering or debiasing SBERT embeddings, in future work.

## REPRODUCIBILITY STATEMENT

To ensure reproducibility, we will release anonymous source code, hyperparameter settings, and data preprocessing scripts as supplementary material upon acceptance. The Million Song Dataset is publicly available. The autoencoder architecture, HDBSCAN parameters, and ARM thresholds are documented in the Appendix (see Table 4). Our implementation of the Association Rule Mining phase uses the `pyfpgrowth` Python library. The SBERT model is a standard pre-trained model (`all-MiniLM-L6-v2`). All random seeds and experimental details are included in the released code to facilitate replication.

## LLM USAGE STATEMENT

This manuscript was prepared with the assistance of a large language model (OpenAI ChatGPT) in editing and improving the writing. Specifically, the LLM was used to refine phrasing and ensure clarity, but all ideas, content, and results presented in this paper are original and authored by the listed researchers. We take full responsibility for the final content. The LLM was not used to generate scientific claims or data; it served only as a writing aid.

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

# A APPENDIX

## A.1 HYPERPARAMETER SETTINGS

Table 4: Hyperparameter settings for the Hy-ClustRec model components.

| Component | Hyperparameter | Value |
|---|---|---|
| Data Preprocessing | User sampling fraction | 30% |
| | Minimum songs per user | 20 |
| | Minimum plays per song | 70 |
| Autoencoder | Latent dimension | 4 |
| | Optimizer | Adam |
| | Loss function | MSE |
| | Training epochs | 30 |
| | Batch size | 128 |
| HDBSCAN | Minimum cluster size | 10 |
| | Metric | Euclidean |
| Hierarchical ARM | Intra-cluster min. support | 5 |
| | Intra-cluster min. confidence | 0.10 |
| | Global min. support | 10 |
| | Global min. confidence | 0.15 |
| Hybrid Scoring | Similarity weight ($w_s$) | 0.5 |
| | Confidence weight ($w_c$) | 4.5 |
| | Membership weight ($w_m$) | 0.5 |
| | SBERT Model | `all-MiniLM-L6-v2` |

## A.2 DETAILED NETWORK ARCHITECTURES

Table 5: Autoencoder architecture for content feature learning.

| Network Part | Layer Type | Output Shape | Activation |
|---|---|---|---|
| Encoder | Input Layer | (None, 2) | – |
| | Dense | (None, 8) | ReLU |
| | Dense (Latent Space) | (None, 4) | ReLU |
| Decoder | Dense | (None, 8) | ReLU |
| | Dense (Reconstruction) | (None, 2) | Linear |

