# OpenReview forum: "Hy-ClustRec: Deep Cluster-Guided Rule Mining for Cold-Start Recommendation"
_ICLR.cc/2026/Conference — ICLR 2026 Conference Desk Rejected Submission_

### Official Review · Reviewer_z8C1 · 2025-10-25

**Soundness:** 2
**Presentation:** 2
**Contribution:** 1
**Rating:** 2
**Confidence:** 5

**Summary:**

Hy-ClustRec:
The authors propose a model called Hy-ClustRec to address the cold-start problem in recommender systems. Hy-ClustRec is a three-stage framework consisting of three phases: deep content-based item clustering, hierarchical association rule mining, and hybrid scoring and re-ranking. The authors perform an experiment on the Million Song dataset and compare their results to two existing baselines.

**Strengths:**

- The problem formulation/ proposed model is concise and clearly elaborated.
- The conceptual overview well summarizes the proposed pipeline of Hy-ClustRec.

**Weaknesses:**

- **Lack of novelty and technical contribution**. The authors simply combines existing models/work into their three-stage framework. How autoencoder, HDBSCAN have been chosen for their scheme haven’t been supported in the manuscript. Given that the pipeline simply combines existing models, are they carefully chosen at least through rigorously comparisons? Hy-CLustRec does not introduce any fundamentally new modeling concepts.
- **Lack of experimental significance**. The authors simply perform an experiment on one dataset. While there are more popular datasets, e.g., Movielens, Amazon, Tiktok, etc, actively being used across multiple papers, the authors simply select one dataset for their experimentations. The efficacy of the proposed scheme cannot be supported through a single experiment with one dataset.  It is also surprising that Hy-ClustRec barely beats the simple approach ItemKNN from early 2000s. This is questionable as Hy-ClustRec is augmenting item-based KNN at phase 3, which raises doubts about the contributions of other factors.
- **Lack of details**. The current draft is missing many important details (see questions). The authors simply report the weights in Eq. 4 and Eq 6. Hy-ClustRec highly rely on a series of heuristics and manually-tuned hyperparameters.
- **Overstated claims**
a. Line 66: “providing a robust and methodologically novel solution for cold-start”, There isn’t any experimental results supporting the robustness.
b. Line 239: “The final dataset presents a significant cold-start challenge”. However, the data preparation seems too general. (line 236-238). Are there any experiments that was performed only on cold items?

(minor) K is being used in two ways: clustering and evaluation. Try to be clear when using K.

(minor) Try to be consistent in style for readability in Equations 7-11.

**Questions:**

Q1. How can the hierarchical ARM mine both fine-grained intra-cluster rules and broad global rules?

Q2. In line 214, it says “are tuned on a validation set”, but according to line 244, there’s only train set and test set with 80:20 split.

Q3. The input dimension of the autoencoder is set to 2 in the Appendix A.2. Further details are required; what is the input into the autoencoder?

Q4. When you use K-means instead of HDBSCAN, how do you fix K for the K-means?

---

### Official Review · Reviewer_pPX8 · 2025-10-29

**Soundness:** 2
**Presentation:** 3
**Contribution:** 1
**Rating:** 2
**Confidence:** 2

**Summary:**

This paper presents a framework designed to address the cold-start problem in RS, where a lack of user interaction data leads to poor recommendation quality. Hy-ClustRec is a three-stage pipeline that combines deep learning for feature representation with hierarchical pattern mining to generate recommendations, especially in data-sparse or cold-start scenarios.

1. Content-Based Item Clustering: a deep autoencoder is used to learn dense, non-linear representations of item content metadata. The autoencoder is trained to minimize the reconstruction error. These learned embeddings are then segmented into meaningful communities using the HDBSCAN clustering algorithm
2. Hierarchical Association Rule Mining (ARM) The framework then mines for co-occurrence patterns at two levels: intra-cluster rulse and global rules
3. Hybrid Scoring and Re-ranking: three scores namely rule-based confidence from ARM, semantic content similarity, and user-cluster affinity are linearly combined to produce a ranking score to be used by KNN recommender.

**Strengths:**

The paper is presented with clarity.

The paper addresses the cold-start problem, a long-standing and critical issue in RS where the absence of interaction data severely limits recommendation quality.

**Weaknesses:**

Limited novelty in individual components:

* stage 1 : Using a deep autoencoder to learn dense, non-linear representations of item content is a standard and widely used approach for feature learning in RS. While effective, this stage functions as a standard preprocessing step rather than a novel contribution to the core methodology. The paper does not seem to introduce any innovations in the autoencoder architecture or training process itself.
* stage 2: Applying a clustering algorithm to item embeddings to segment the catalog is also a common strategy for tasks like candidate generation or understanding item communities. The choice of HDBSCAN is frequnently seen in prior publications. The act of clustering item features is not in itself a novel idea.

*  The use of manually tuned weights (w_c​​, w_s, w_m​) is a heuristic approach. It lacks a "principled" mechanism where the optimal combination of these features could be learned directly from the data. Modern systems often use model-based approach in the final stage to learn the complex, non-linear interactions between different features (the ranker). A simple linear combination is less expressive and may not be optimal.

*  While Association Rule Mining (ARM) is effective, its computational complexity can be a concern for large-scale industrial systems. The paper proposes mining "Global Rules" from the entire dataset, which could become a significant bottleneck as the number of users and items grows into the millions or billions. The paper does not address the scalability of this hierarchical rule mining process, which is a critical consideration for real-world deployment.

**Questions:**

* The sequential, multi-stage pipeline (autoencoder -> clustering -> rule mining) creates a risk of error propagation, where suboptimal performance in an early stage can negatively impact all subsequent stages.
Question: How robust is the framework to noise or suboptimal outputs from the earlier stages? For example, if the autoencoder fails to produce high-quality embeddings, how does this affect the final recommendations? Do you have any measure or alerts setup in the system to detect that?

* Could you discuss the scalability of the global ARM stage? As the number of users and items in a real-world system grows, mining rules from the entire dataset can become computationally prohibitive. What was the runtime performance on the Million Song Dataset subset, and how do you see this approach scaling to industrial-level datasets?

* The final ranking is determined by a hybrid scoring function that combines rule confidence, semantic similarity, and a user-cluster affinity score, augmented with a KNN model. This appears to be a linear combination of several heuristic scores.  Could you elaborate on the methodology used to determine the weights? Was this done via a simple grid search, and if so, how sensitive was the model's performance to these hyperparameters?

---

### Official Review · Reviewer_MJ2E · 2025-11-02

**Soundness:** 2
**Presentation:** 3
**Contribution:** 1
**Rating:** 0
**Confidence:** 5

**Summary:**

The paper proposes a pipelined method for cold-start recommendation. An autoencoder maps item metadata to a low-dimensional embeddings; HDBSCAN then partitions items into communities and marks outliers; FP-Growth is applied both within cluster and globally to extract rules; and get a hybrid re-ranking step: candidate items are scored with a weighted sum of (i) rule confidence, (ii) SBERT similarity between a candidate's text embedding and the user's mean profile embedding, and (iii) a user-cluster membership term. A non-truncated item-based KNN score is then blended via per user min-max normalization. Experiments are ran on a subset of MSD dataset with filtering: 30% user sampling, more than 20 interactions per user, and more than 70 plays per song. The authors report top-K metrics and compare against ItemKNN and a pipeline baseline CFCAI.

Overall it is my opinion that this contribution is incremental; shows small gains over a weak baseline, and importantly, lacks proper cold-start evaluation.

**Strengths:**

(+) Clear modular pipeline: What is described would be a reasonable design point for transparency and practical deployment.
(+) Ablation study suggests each component contributes to the overall performance of the method.

**Weaknesses:**

(-) The Evaluation does not match the 'cold-start' claim. The paper filters out low-frequency items, which removes tail items where cold-start is most severe. The split is a random per-user 80/20 interaction split, not a user-level or item-level cold-start protocol. As written, the experiments primarily measure standard top-K retrieval with moderately active users and relatively popular items, not cold-start robustness. Frankly, the filtering employed in the paper, is pretty typical pruning of the user-item interaction graph one finds in methods tackling standard top-n recommendation in the literature, without any cold-start emphasis.
(-) Comparison against cold-start focused baselines is missing. RecSys literature contains a large number of papers with an emphasis on cold-start. Also, it is understood in this space that graph-based and so-called latent-space methods are inherently better at addressing the cold-start problem. However the paper does not compare against these methods. From the latter category EASE-r, RecWalk, SLIM, Mult-VAE, LightGCN are standard points of comparison, and a good place to start. Current experiments make it hard to place the contribution against the state-of-the-art.

**Questions:**

- Eq 5 defines M as item x user with M_{iu} = 1 iff user u played item i. It is not stated that M is built only from training interactions. Is it? If not S has access to "future information", artificially inflating results. Please clarify.
- Appendix Table 5 shows an autoencoder input of only 2 features ("(None, 2)") with a latent size of 4, the claim "dense, non-linear content representation" is weak; an overcomplete autoencoder on 2D input offers little representational power and may even be worse than simple one-hot/embedding lookups. Please explain.

---

### Note · Program_Chairs · 2026-01-17
**Submission Desk Rejected by Program Chairs**

The following references in this submission do not refer to real documents and/or have major errors in bibliographic information:

 Thabit S. Al-Obeidi, Baidaa A. Mahdi, and Falih M. Al-Naima. A brief overview of music recommender systems. Journal of Engineering and Applied Sciences, 13(15):6156-6167, 2018.

Lexiang Chen, Jia Chen, Zhiyong Chen, and Guan Liu. A hybrid model for recommendation with feedback data by fusing latent dirichlet allocation and latent factor model. Frontiers in Applied Mathematics and Statistics, 5, 2019. ISSN 2297-4687. doi: 10.3389/fams.2019.00044.